# Applications of Nonconventional Green Extraction Technologies in Process Industries: Challenges, Limitations and Perspectives

**Gertrude Fomo \*, Tafirenyika Nyamayaro Madzimbamuto and Tunde Victor Ojumu** 

Department of Chemical Engineering, Faculty of Engineering & the Built Environment, Cape Peninsula University of Technology, Bellville Campus, Bellville 7553, South Africa; MadzimbamutoT@cput.ac.za (T.N.M.); OjumuT@cput.ac.za (T.V.O.)
\* Correspondence: FomoG@cput.ac.za or fomogertrude@gmail.com; Tel.: +27-83-758-6805

**Abstract:** This study reviewed five different nonconventional technologies which are aligned with green concepts of product recovery from raw materials on industrial scale, with minimal energy consumption and chemical use. Namely, this study reviewed supercritical fluid extraction (SCFE), pressurized liquid extraction (PLE), microwave-assisted extraction (MAE), ultrasound extraction (UAE) and pulsed electric fields extraction (PEFE). This paper provides an overview of relevant innovative work done in process industries on different plant matrices for functional value-added compounds and byproduct production. A comparison of the five extraction methods showed the supercritical $CO_2$ (SC-$CO_2$) process to be more reliable despite some limitations and challenges in terms of extraction yield and solubility of some bioactive compounds when applied in processing industries. However, these challenges can be solved by using ionic liquids as a trainer or cosolvent to supercritical $CO_2$ during the extraction process. The choice of ionic liquid over organic solvents used to enhance extraction yield and solubility is based on properties such as hydrophobicity, polarity and selectivity in addition to a safe environment.

**Keywords:** green extraction technology; value-added compounds; process industries; ionic liquids

## 1. Introduction

Recently, considerable emphasis has been placed on the reduction of carbon footprint for the development of smart cities worldwide. Chemical process industries can play an important role in achieving this objective by adopting technology which is environmentally sensitive using green solvents in separation and extraction processes, referred to as green technology. Green solvents and sustainable technologies are integrated into the extraction and separation of large numbers of bioactive products from biomass conversion. The new paradigm shift in the usage of bioactive products in multiple industries like pharmacy, food, refinery and textile suggests on one hand the acceptance of the need for healthy lifestyles and on the other hand, the need for chemical industries to use safe and appropriate techniques for extraction from biomass or fermentation broths [1–3]. In view of the foregoing, many green extraction and separation techniques including non-conventional as well as conventional methods ... have been reported [4–6].

Nonconventional techniques including supercritical fluid extraction (SCFE), pressurized liquid extraction (PLE), microwave-assisted extraction (MAE), ultrasound extraction (UAE) and pulsed electric fields extraction (PEFE) have been reported in different process industries [1,7,8]. The above mentioned extraction methods are advantageous compared to conventional techniques [9]. Indeed, these techniques can operate without light and oxygen, at elevated temperature or pressure,

combined with low organic solvent consumption and extraction time. Long extraction time, high solvent cost, low selectivity and thermal decomposition of thermos-labile compounds are major challenges of conventional extraction [10]. These limitations can be removed using nonconventional extraction techniques.

The processing industry is regarded as one of the fastest growing sectors today with an estimated product growth rate of 8.6% per year [11]. The advancement in processing has been instrumental in the growth of pharmaceuticals, chemicals, food, textiles and refinery industries [12]. These industries are networks of equipment covering a number of combined technologies leading to the sustainable transformation of biomass for the concomitant production of biofuels, chemicals, natural products, food supplements or active compounds, preferably those which add value [13]. In this review, some of the nonconventional extraction techniques employed in processing industries are discussed and compared with the view to their merits and demerits, especially with respect to their application to a typical scale-up process and their economics. The view is to discuss the comparative advantage of high pressure technology with respect to SCFE methods especially as it applies to extraction or fractionation of biological/bioactive compounds and to highlight its comparative advantage when combined with ionic liquid as additional solvents to achieve more efficient extraction and fractionation.

## 2. Process Industries

Typical process industry raw material feedstocks are processed in a production plant which may consist of different unit operations working in continuous and/or batch mode. This definition is schematized in Figure 1, which explains the complex production chain that characterizes significant features with regard to products and processes developed in the processing industry. The greener nature plus high efficiency of nonconventional technologies have been shown to make them superior to conventional methods, especially with respect to product quality and integrity. While some industries still rely solely on the conventional technologies, the following section will discuss some selected nonconventional techniques used in the process industry that are relevant to refinery, food, health and textile. Details of these conventional technologies have been reviewed elsewhere [14]. Table 1 describes products that can be obtained from various feedstocks (raw materials) using nonconventional techniques reviewed in this paper in process industries with a focus on food and pharmaceutical industries. This shows that more interest has been given to food and pharmaceutical industries as compared to refinery, textile, and chemical industries using nonconventional techniques.

**Figure 1.** A schematic production process in industry.

**Table 1.** Some processing target ingredients and their sources in different industrial sectors reviewed in this work.

| Industrial Sectors | Sources | Target Ingredients | Refs. |
|---|---|---|---|
| Refinery | - Oil sands, oil shale rock, lignite, tire, rubber | - Crude oil, diesel fuel, kerogen, hydrogen, methane | [15–17] |
| | - Tar sands | - Bitumen | [18] |
| | - Cannabis seeds | - Oil | [19] |
| | - Palm kernels | - Palm kernel oil | [20–22] |
| | - Heavy petroleum feedstocks | - Vanadium, nickel and iron | [23] |
| pharmaceutical | - Tinctures of sage, valerian | - Cineole, borneol, mixture of α- and β-thujones | [24] |
| | - Penggan (*Citrus reticulata*) peel | - Hesperidin | [25] |
| | - Grapes | - Tartaric acid | [26] |
| | - Vegetal sources | - Flavonoids | [27] |
| | - Bovine meat, porcine meat, poultry meat, sea bream fish, trout fish | - Antibiotics (Erythromycin A, Josamycin, Roxithromycin, Spiramycin, Tilmicosin, Troleandomycin, Tylosin) | [28] |
| | - Meat | - Tetracyclines (tetracycline, chlorotetracycline, oxytetracycline and doxycycline), sulfonamides | [29,30] |
| | -Sewage sludge | - Macrolides | [31] |
| | -Fish | - Paroxetine and fluoxetine | [32] |
| | - *Chlorella vulgaris* | - Carotenoid and chlorophylls | [33] |
| | - Soil | - Estrone (E1), 17β-estradiol (E2), 17α-ethinylestradiol (EE2), estriol (E3) | [34,35] |
| | - Red pepper | - Vitamin E and provitamin A | [36] |
| Food | - olive leaves/fruit | -Sugars (polysaccharides), protein, fatty acids, pigments, and polyphenols, polyalcohols, lipids and pectins, tyrosol, hydroxybenzoic acid, cinnamic acid, hydroxytyrosol, caffeic acid, syringic acid, elenolic acid, chlorogenic acid, ligstroside, oleuropein, verbacoside, tocopherols, p-coumaric acid, vanillic acid, elenolic acid, catechol and rutin | [37–40] |
| | - Carrrot, grapes, black tea, apple ginger | - Beta-carotene, polyphenol, tartaric acid, phenol, antioxidants, anthocyanins, polyphenol, aroma components, polyphenols, gingerol | [9,41,42] |
| | - Ginseng roots, | - Ginseng saponins | [43] |
| | - Caraway | - Caryone and limonene | [9] |
| | - *Amaranthus caudatus* seeds | - Vitamin E | [9] |
| | - Adlay seed | - (*Coix lachrymal-jobi L. var. Adlay*) oil and coixenolide | [44] |
| | - Egg yolk | - lutein | [45] |
| | - Garlic | - Essential oil: organo-sulfur compounds | [46,47] |
| | - Citrus Flowers and honey | - Linalool | [28] |
| | - Peppermint leaves | - Menthol | [48] |
| | - Tomatoes | - Carotenoids (all-trans-lycopene, b-carotene) | [49] |

**Table 1.** *Cont.*

| Industrial Sectors | Sources | Target Ingredients | Refs. |
|---|---|---|---|
| | - Vegetal sources | - Phenolic acid, tannins, stilbenes | [50] |
| | - Crude palm oil and soybean oil | - Tocotrienols and tocopherols | [51] |
| | - Olive mill wastewater, sea buckborn, berry pomace, mango peel, white grape pomace | - Phenols | [52] |
| | - Soybean | Isoflavones | [53] |
| | -Artichoke, blueberry, grapes and grape seeds, flaxseed hulls, orange peel, and tomato | - Phenolic compounds, flavonoids, anthocyanins, carotenoids | [54] |
| | - Orange juice and pasteurized juices | - Vitamin C | [55] |
| | - Fed cabbage | - Anthocyanin | [52] |
| | - Crude palm and soybean oil | - Tocotrienols and tocopherols | [56] |
| Textile | -Dye (turmeric, henna) | - Turmeric and henna extracts | [57] |
| | -Mordants (harda, tamarind seed coat) | - Harda and tamarind extracts | [58] |
| | - Red Beetroot (*Beta vulgaris*) | - Cyanins and betanin (red) and Betaxanthins. (Yellow) | [52] |
| | - Lac (*Kerria lacca*) | - Lacciac acid | [59] |
| | -*Dastylopius coccus, kermes licis, kerria lacca,* | - Cochineal (red), Kermes (red), lac | [60] |
| | -Artichoke, red cabbage, blueberry, grape seeds, flaxseed hulls, orange peel, tomato | - Phenolic compounds, flavonoids, anthocyanins, carotenoids | [51] |
| | - *Ricinus communis* leaves | -Natural dye | [61] |
| Chemical | - Grapes | -Tartaric acid | [35] |
| | - Corn starch or corn pericarp | -Carbohydrates (glucose, xylose and arabinose) | [62] |

## 3. Nonconventional Extraction Methods in Process Industries

### 3.1. Ultrasound-Assisted Extraction (UAE)

UAE is a suitable, cost-effective, faster kinetic and efficient nonconventional technique in solid-liquid extraction operating at 20 kHz to 100 MHz, for heat sensitive compounds with minimal damage [63]. UAE technique is a relatively novel, environmentally friendly and economically viable alternative to conventional techniques applied in food, environmental, pharmaceutical, textile and chemical industries. The main improvement in works published on medicinal plant extraction using ultrasound is found to be efficiency and short time of extraction. Valachovic et al. [24] extracted tinctures of sage (*Salvia officinalis* L.) and valerian (*Valeriana officinalis* L.) with high antimicrobial activities using UAE at 20 kHz (600 W) for 6 h with ethanol/water. Yaqin Ma et al. produced hesperidin from Penggan (Citrus reticulata) peel at 60 kHz frequency and temperature of 40 °C for 1h in methanol [25]. In the pharmaceutical industry, the active zone and distribution of ultrasonic power are challenges that should be considered. The application of UAE technology in food processing is of interest to increase the extraction yield of polyphenolics, anthocyanins, aromatic compounds, polysaccharides and oils from plant and animal materials. Wu et al. [44] in 2001 and Chemat et al. [45] in 2004 used UAE at lower temperatures to extract ginseng saponins from ginseng roots, and caryone and limonene from caraway respectively. As egg yolk is an important source of lutein in some foods [64], Xiaohua et al. [43] extracted lutein from egg yolk by using sonication. From this study, it was found that depolymerization can lead to a decrease in the degree of acetylation which can limit this technique in food processing. Xiaohua et al. [43] obtained a large amount of luetin using UAE combined with saponificated organic solvent. Essential oil has been extracted from peppermint leaves [51], artemisia [65] and lavender [63] and from garlic [66] and citrus flowers [52] using UAE. For textile industry application, Sheikh et al. [55] extracted natural dyes (turmeric, henna) and mordants (harda, tamarind seed coat) for textile materials using UAE in water in comparison with conventional methods. Moreover, Kamel et al. [60] in 2004 studied the effect of pH, salt concentration, ultrasonic power and dyeing time during the extraction of lacciac acid dye from lac (dye material of animal origin) in distilled water for dyeing wool fabric. Due to various mechanisms with cavitation in liquid, ultrasound has been beneficially applied in chemical processes. Presently, despite the limitation of UAE due to low efficiency and high operation costs, it is a promising extraction method for added-value pharmaceuticals, food and chemical ingredients on an industrial scale [67–71]. One of the challenges of UAE in the chemical industry is to obtain effective mass transfer with suitable mixing in order to create a strong interaction between the two immiscible phases.

### 3.2. Microwave-Assisted Extraction (MAE)

Microwaves are electromagnetic fields in the range of 300 MHz to 300 GHz. Microwave heating affects the frequency, power and speed, temperature, mass of food, water content, density, physical geometry, thermal properties, electrical conductivity and dielectric properties of molecules. No heating occurs when the frequency is less than 2450 MHz and the electrical component changes at a much lower speed. MAE is commonly used in organic and organometallic compound extraction; it utilizes water or alcohols as solvent at elevated temperatures and controlled pressure conditions [68]. However, for refinery application, numbers of reports have successfully investigated the extraction and separation of oil sands or oil shale. MAE was used to separate two samples of oil sands at 915 MHz, irradiated for 5 and 9 min at 500 W and 1500 W separately and reached final temperatures of 315 °C and 142 °C respectively [18]. These samples showed three distinct layers with the bottom layer mostly sand but also containing other solids; the second layer consisted of a yellowish solution, accounting for all the water and other impurities in the oil sand; and the top layer was black, viscous oil. Moreover, Bosisio et al. [72] used MAE to extract oil sands at room temperature in a quartz reactor with an incident microwave power of 100 W at 2450 MHz frequency. In addition, Balint et al. [73] described the irradiation of oil sand, oil shale rock and lignite samples in a pressure vessel with gaseous

or liquefied carbon dioxide and other gaseous or vapor hydrocarbon solvents. In pharmaceutical industry application, various studies have reviewed the use of MAE, some of which this work reviews. Ergosterol was produced from hyphae and spores using SCFE with lower yield compared with that of MAE [74]. Norproxifen materials as model medicines are the main applications in which the MAE technique is used. Carbamazepine, the most commonly employed pharmaceutical in the treatment of epilepsy and neuropathic pain, was recovered from wastewater sludge using MAE at 1200 W power and 110 °C for 10 min [75]. MAE has been effectively employed to produce many classes of antibiotics like levofloxacin, norfloxacin, ciprofloxacin, enfloxacin, fluoroquinolones, tetracyclines from soil [76], sewage and wastewater sludge [77,78], compost [79] and aquifer sediments [74]. In the food industry, Koh et al. [80] investigated the recovery of pectin from jackfruit rinds at 90 °C at 450 W power for 10 min. On the other hand, the extraction of phenolic compounds from olive leaves and Melissa using MAE was reported by Rafiee et al. [81] and Ince et al. [82]. Rafiee et al. recovered phenolic compounds from olive leaves with different solvents at 15 min while Ince et al. focused on the extraction time and solid-to-solvent ratio for recovery of total phenolic compounds. Among the solvents used, ethanol was the most efficient where acetone was the least. In textile industry application, Sinha et al. [83] evaluated the effect of extraction time, pH of solution and the amount of pomegranate rind produced from brown dry rind of pomegranate using MAE for textiles. The effect of water and ethanol solvent at different temperatures with different material to solvent ratio and microwave power was studied for the extraction of natural tannins and flavonoid dyes from coleus atropurpureus leaves [58]. Raza et al. [84] extracted natural colorant for textile industry use from munj sweet cane by optimizing the temperature, time, fabric to extract ratio, salt type and salt concentration. In the chemical industry, few studies have been done on extraction using MAE. In 2010, Yoshida et al. [85] optimized the heating temperature, heating time and solid to liquid ratio when extracting carbohydrate compounds from corn pericarp with water as a solvent. In all the reviewed studies aforementioned, polarity or volatility of compounds/solvent may reduce the efficiency of the MAE, and more importantly, the high temperature requirement of this technology may preclude its application for extraction of thermolabile compounds such as can be found in bioactive materials and/or in compounds of pharmaceutical relevance. In addition, scaling of MAE system for industrial application is still in its infancy and largely limited to laboratory scale.

### 3.3. Pressurized Liquid Extraction (PLE)

Several bioactive compounds are extracted from different plants using PLE which is an alternative method used thanks to short time operation extraction (<30 min), inert environment under high pressure (<20 MPa) and temperatures (25–200 °C) and low solvent consumption [86]. This technique can be applied in different sectors like the pharmaceutical and food industries. In pharmaceutical application, Blasco et al. [87] evaluated the temperature, pressure, treatment of sand, static time, cell size, number of extraction cycles and flush volume for the extraction of four tetracyclines (tetracycline, chlorotetracycline, oxytetracycline and doxycycline) from different types of meat with water as solvent. Pharmaceutical compounds like estrogens are widely used products because of their contraceptive activity. Indeed, different works have reported on the production of estrogens such as estrone (E1), 17β-estradiol (E2), 17α-ethinylestradiol (EE2) and estriol (E3) from soil using PLE [87]. Other studies investigated the extraction of other pharmaceutical compounds such as sulfonamides in meat [88] and paroxetine and fluoxetine in fish [31]. In addition, with regard to the food industry, Kwang et al. [89] optimized temperature and time to extract carotenoids and chlorophylls from the green microalga *Chlorella vulgaris* using PLE. On the other hand, reference [90] explored the extraction of polyphenols from the needles of *Pinus taiwanensis* and *Pinus morrisonicola* which involved enzymatic hydrolysis with a combination of many enzymes (cellulase, hemicellulase, pectinase and protease). Much interest has been given to essential oils from herbal plants in the food, pharmaceutical and cosmetic industries due to their antimicrobial, antioxidant and antifungal activities, aroma-active constituents and flavor properties [91–93].

### 3.4. Pulsed Electric Fields Extraction (PEFE)

The principle of PEFE is to induce the electroporation of the cell membrane, thereby increasing the extraction yield. The pulse amplitude ranges from 100 to 300 V/cm to 20 to 80 kV/cm. Solvent as water and agri-solvents (ethanol and methyl esters of fatty acids from vegetable oils) are used in PEFE, hence this technique is considered a green extraction technique producing extracts of high quality and purity [94]. PEFE is mostly applied in food preservative from plant food materials, food wastes and byproducts [95,96]. This technique can be applied in the biorefinery, pharmaceutical and food industries. In the biorefinery industry, the extraction of intracellular compounds is the most interesting feature of PEF treatment. For example, the extraction of sucrose from sugar beet, colorants from grapes and red beet, polyphenols from green biomass and lipids from microalgae use the pulsed electric fields method [97]. Haji-Moradkhani et al. [19] evaluated the speed of pulsing oil on the physicochemical properties of cannabis oil extracted from cannabis seeds. Leong et al. [98] produced anthocyanins from grape juice after PEF treatment at 20 mS pulse with 50 Hz under an electric field strength of 1.5 kV/cm. This technique was shown to be efficient by providing pure extracts with the ability to protect cells from oxidative stress. Moubarik et al. [99] reported that fennel (*Foeniculum vulgare*) extracts with valuable antioxidants and medicinal properties have also been obtained using PEFE and the extraction yield was enhanced. PEF treatment applied to artichoke, blueberry, grapes and grape seeds, flaxseed hulls, orange peel and tomato enhance the yield of phenolic acid, flavonoids, anthocyanins and carotenoid compounds [52]. The extraction of polyphenols and flavonoid derivatives from orange peel using PEFE has been reported [48]. Moreover, Segovia et al. [100] investigated the isolation of polyphenols from *Borago officinalis* leaves using PEF treatment. The polyphenol and oxygen radical absorption capacity (ORAC) values were increased between 1.3% and 6.6% and from 2.0% to 13.7%, respectively. Jaeger et al. [101] used the PEF technique for extraction of apple juice. Elez-Martinez and Martín-Belloso extracted vitamin C from orange juice and the content of vitamin C was compared with that in conventionally pasteurized juices which showed higher content in PEF-treated orange juice than in pasteurized juice [54].

### 3.5. Supercritical $CO_2$ (SC-$CO_2$) Extraction

Supercritical extraction is based on the change in temperature and pressure which transform the gas used in supercritical fluid. Supercritical fluid (SCF) is fluid whose temperature and pressure values are above the critical temperature (Tc) and critical pressure (Pc) values. Tc and Pc values are different for each supercritical fluid. $CO_2$, $N_2O$, $NH_3$, $H_2O$, $CHF_3$ and $SF_6$ compounds have been considered as supercritical fluids in the SCF process [102,103]. For example, Tc and Pc values for these supercritical fluids are presented in Table 2. $CO_2$ is the most commonly used SCF because of its low critical temperature and nonexplosive character; it is safe and inexpensive and has important qualities for pharmaceutical application. Controlled increase in temperature and pressure values brings these two parameters together at a common point called the "critical point." Thus, Tc and Pc values are the temperature and pressure values possessed at the critical point (see Figure 2). When this point is reached, the solvent is described as SCF.

**Table 2.** Critical conditions of some fluids used in supercritical fluid (SCF) extraction and separation.

|          | $CO_2$ | $H_2O$ | $N_2O$ | $CHF_3$ | $SF_6$ | $NH_3$ |
|----------|--------|--------|--------|---------|--------|--------|
| Tc (K)   | 304.2  | 646.8  | 309.6  | 299.1   | 318.7  | 405.59 |
| Pc (MPa) | 7.37   | 22.06  | 7.24   | 4.83    | 3.76   | 11.28  |

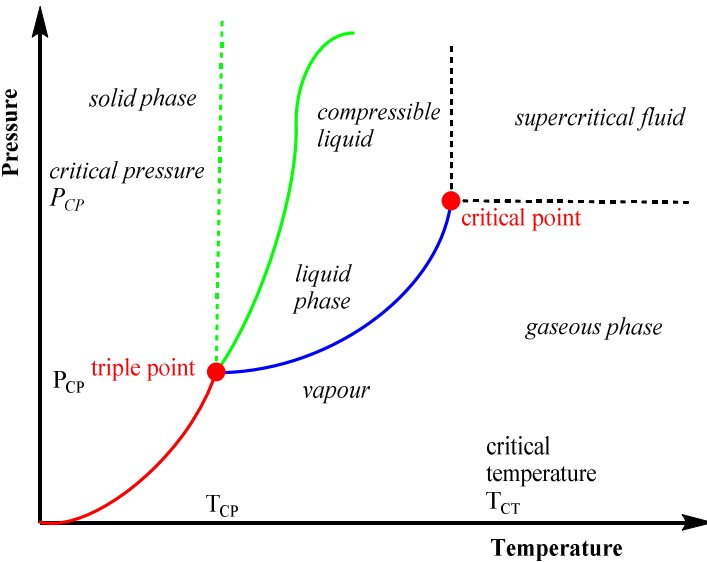

**Figure 2.** Diagram of critical temperature, pressure and critical point.

Supercritical $CO_2$ (SC-$CO_2$) extraction has been applied in various sectors such as the food, chemical and pharmaceutical industries [104,105]. The main limitation remains its high cost and the difficulty of performing continuous extractions [106]. The SC-$CO_2$ process in the oil refinery industry is used to deasphalt the oil and heavy residues by reducing the solvent ratio when removing raw materials and improving the efficiency of the process. Magomedov et al. [23] investigated the demetallization of heavy petroleum feedstocks (HPF) using SFE for the extraction of vanadium, nickel and iron. In addition, many other researchers have investigated the extraction and fractionation of palm kernel oil from palm kernels using SCFE [19–22]. The yield of palm kernel oil was shown to increase with pressure (34.5 to 48.3 MPa at 353.2 K). At lower pressure (20.7 to 27.6 MPa), lower amounts of shorter chain triglycerides (C8-C14) were obtained. Fatty acid constituents with long chain (C16-C18) were extracted at higher pressures from 34.5 to 48.3 MPa. At higher temperature, the extracted palm kernel oil was found to be superior in terms of the constituent of fatty acids in triglycerides compared to that obtained from Soxhlet extraction using hexane. Pharmaceutical/cosmetic active compounds can be obtained from either natural or synthetic and polymeric patches or implants and cosmetic lacquers using the SCF process. Romo-Hualde et al. [107] in 2012 evaluated various parameters that can affect the extraction yield using $CO_2$ as the supercritical fluid. These authors found that the highest yield of vitamin E (97%) and provitamin A (68.1%) in red pepper (*Capsicum annum* L.) can be obtained at a temperature of 60 °C at 24 Pa and using a particle size of 0.2–0.5 mm. In food applications, many important food ingredients from the health point of view have been extracted using SC-$CO_2$ process by many researchers [108]. Products produced by SCF technology such as flavored orange juice, vitamin additives, dealcoholized wine/beer, defatted meat, defatted potato chips and spice extracts are commonly found on our tables [109]. Moreover, Guedes et al. [110] used supercritical fluid to produce carotenoids and a, b and c chlorophylls from *Microalga Scenedesmus obliquus* for subsequent use in food processing. The highest yields were obtained at 250 bars. The most suitable temperature to obtain the highest yield for chlorophyll was 40 °C and for carotenoids was 60 °C. The supercritical fluid process is also used in the refinery industry for food ingredient production such as colorants, antioxidant preservatives, texture agents and low-fat products. In the textile industry, the phase behavior of the system consisting of solids and gas should be well described when working with supercritical systems in dyeing [111]. Saus et al. [111] reported in situ results on the dyeing of poly (ethylene terephthalate) and other synthetic materials with factors influencing dye uptake and levelness. In addition, Sanchez-Sanchez et al. [112] impregnated a polyester textile with mango leaf extracts (MLE) in the mixture of $CO_2$/methanol (50%) at 120 bar and 100 °C and the extracts presented antioxidant, bacteriostatic and bactericidal activities. SCFs are used in some chemical reactions (ammonolysis,

alkylation, polymerization, field crafts, hydroformylation, hydrogenation, inorganic catalytic processes, metathesis and oxidation) that have already been implemented at industrial scale to obtain added-value products [113]. Examples are the ammonolysis reaction of ester to amides in supercritical ammonia, the asymmetric hydrogenation in SC-$CO_2$ and also the synthesis of D, L-$\alpha$ Tocopherol in SC-$CO_2$ and in SC-$N_2O$ [114]. In addition, the hydrogenation reaction of organic compounds in SCFs was implemented at industrial scale [115] by Thomas Swan and Co Ltd Company in 2000.

## 4. Comparison of SCFE with Other Extraction Methods

Table 3 compares the green extraction methods considered in this review in terms of their operational conditions. Unlike SCFE (that uses SC-$CO_2$), all these techniques in addition to water, ethanol and methanol use other organic solvents which compromise the greener nature of the technique. When comparing energy consumption, SCFE process requires less energy than other techniques. These make SCFE more economically suitable for research and industrial applications. Even though UAE, MAE, PLE and PEFE methods are widely applied in extraction fractionation and separation processes, they present certain limitations such as high energy costs, low selectivity and large quantities of solvent waste [116] which is not the case with SCFE. Besides that, MAE and PLE techniques are not suitable for thermolabile compounds due to high temperature applied; the advantage of the SCF $CO_2$ process is that it does not destroy thermolabile compounds at critical temperature and pressure. Despite all the advantages of SCFE, this technique also has certain limitations when extracting or fractionating separating polar compounds which are due to the nonpolar character of $CO_2$. However, many researchers have shown that the use of a polar solvent as modifier or cosolvent will overcome the challenge [117].

The addition of polar solvents such as acetone, *n*-heptane, toluene or mixtures of toluene and *n*-heptane with acetone and ethanol, methanol, ethanol, acetonitrile, ethyl acetate, toluene and *o*-xylene have been used to increase the solvent power of supercritical fluids [118]. Even though these polar solvents improve the extraction yield and the viscosity of the $CO_2$, they compromise the green qualities of SCF technology. Therefore, the alternative way to increase the polarity of the $CO_2$, yield of extraction and extraction time without affecting the environmentally friendly nature of this SC-$CO_2$ technique could be the use of ionic liquids as cosolvents or modifiers. A number of theoretical and experimental works have already been reported to prove the concept of ionic liquids in green extraction and separation technology (MAE, UAE, PLE, PEFE) in processing or bioprocessing industries [119,120]. However, the application of ionic liquids as cosolvents or entrainers or modifiers for the extraction and recovering of various compounds with high concentration using SC-$CO_2$ technique is rare. As presented in some studies, ionic liquids are 100% soluble in SC-$CO_2$ so the recovery of value-added compounds using SCFE with ionic liquids as modifier will be very easy.

**Table 3.** Summary of nonconventional (green) extraction methods considered in this review.

| Conditions | UAE | MAE | PLE | PEFE | SCFE | Comments |
|---|---|---|---|---|---|---|
| Extraction solvent | Hexane, water, Dichloro-methane, Acetone, ethanol | Hexane, water, Dichloro-methane, Acetone, ethanol | $H_2O$/EtOH, Ethanol | EtOAc/MeOH/$H_2O$, $CO_2$/MeOH, olive oil | SC-$CO_2$, SC-$CO_2$/MeOH SC-$CO_2$/EtOH | • $CO_2$ suitable for bioactive compounds<br>• $CO_2$ density $CO_2$ can be controlled<br>• $CO_2$ has low viscosity and high diffusivity<br>• MAE requires large solvent volume<br>• In SCFE, $CO_2$ has difficulty in dissolving compounds with high molar mass like carotenoids, triacyl-glycerides . . . . |
| Extraction temperature | 10–70 °C | 80–350 °C | 50–190 °C | <100 °C | 30–100 °C | • MAE is applied at elevated temperatures<br>• PLE, MAE not suitable for thermos labile compounds<br>• With SCFE, low critical temperature and low reactivity of $CO_2$.<br>• SFE performed at low temperatures, ideal for thermally labile compounds |
| Extraction time | 10–90 min | 3–30 min | <1 h | <1 h | 10–60 min | • with SCFE, few tens of minutes for extraction time<br>• MAE has the limitations of requiring longer extraction time |
| Power Amount | 48–600 W | 180–1200 W | Moderate | 30 kW | Moderate | • In SFE, changing pressure/temperature changes solvation power leading to high selectivity |
| Cost effective | High cost | Low cost | Low cost | Low cost | Relatively low | • SCFE: operating cost (cleaning and maintenance) underestimated<br>• PEFE, the electricity cost about $0.33 per metric ton<br>• PLE, high manufacturing cost US$29.2/kg extract compared to PEFE<br><br>For MAE, low production cost |
| Industrial sectors | - Food<br>- Pharmaceutica<br>- Chemical<br>- Textile | - Refinery<br>- Food<br>- Pharmaceutical<br>- Chemical<br>- Textile | - Food<br><br>Pharmaceutical | - Refinery<br>- Food<br>- Pharmaceutical | - Refinery<br>- Food<br>- Pharmaceutical<br>- Chemical | • UAE, MAE, promising in pharmaceutical, food and chemical industries<br>• MAE, mostly applied in refinery, pharmaceutical and textile industries<br>• PLE widely applied in food and pharmaceutical industry<br>• PEFE applied in food and pharmaceutical<br><br>SCFE widely used in food, textile, refinery, pharmaceutical and chemical industries |

## 5. Ionic Liquids as Green Solvent

Ionic liquids (ILs) are liquid salts at temperatures <100 °C, [121], constituted of asymmetrical organic cations, organic or inorganic anions and are described as solvents designed with adjustable properties such as thermophysical, biodegradability, toxicity and hydrophobicity features [122]. Many protic and aprotic ILs such as ethylammonium nitrate, $(C_2H_5)NH_3^+·NO_3^-)$ and 1-butyl-3-methylimidazolium have been synthesized since 1914 [123,124]. More attention has been given to aprotic ILs for their excellent properties such as thermal stability over a broad temperature range, electrochemical stability, nonflammability and low volatility and high ionic conductivity [125]. These properties make them potentially effective solvent materials for organic synthesis, extraction and separation processes, catalysis, electrochemistry and also as a good stabilizing agent for proteins, enzymes and nucleic acids [126]. The thermal stability, solubility and viscosity which are essential properties of ILs for their application in supercritical fluid extraction and separation of bioactive added-value compounds recovered from biomass have been well investigated in a number of works [127,128]. In addition, the lack or extremely low vapor pressure of ILs at room temperature is a very significant feature leading to their categorization as green solvents, and this feature has been explored in many applications [129,130].

### 5.1. Ionic Liquid/SC-$CO_2$ Extraction or Fractionation

Much research focusing on ILs has reported on their application in chemical and processing industries; however, their suitability in SCF processes may be attributed to their thermal stability, solubility and viscosity. Their application in the recovery of added-value compounds from biomass material has been well investigated and studied in a number of published works [131,132]. The ILs/SC-$CO_2$ system is relatively new and unexplored at least in process industries. The mechanism is such that organic salts in liquid form interact selectively with polar and nonpolar compounds using π-stacking interactions, hydrogen bonding, ion exchange and hydrophobic interactions, and due to the ionic interactions, the quality and efficiency of the extraction is very high [133]. In metallurgical application, Mekki et al. [134] extracted trivalent lanthanum and europium from nitric acid solution using a three-step extraction system of water/ionic liquid/SC-$CO_2$. This work used the imidazolium-based (1-butyl-3-methylimidazolium (BMIM)) ionic liquid, with bis(trifluoromethylsulfonyl)-imide $(Tf_2N^-=(CF_3SO_2)_2N^-)$ as counter anions with low complexing abilities. Shaofen Wang et al. [135] on the other hand extracted uranyl ions $[UO_2]^{2+}$ from nitric acid solution with imidazolium-based ionic liquid with tri-*n*-butyl phosphate using SC-$CO_2$. The uranyl complex recovered in the SC-$CO_2$ phase was identified to be $[UO_2(NO_3)_2-(TBP)_2]$. Figure 3 shows the schematic diagram of the two biphasic system for extraction process. From the literature up to date, no work has been reported using an IL/SC-$CO_2$ mixture for extraction of bioactive products in refinery, pharmaceutical, food, textile and chemical industries. However, from work done in these industrial sectors using other green techniques than SCF, with VLE data and phase behavior of IL/SC-$CO_2$ physical and critical properties and with IL/SC-$CO_2$ interaction studies [136], we can therefore say in this review that the extraction or fractionation of value-added products using IL/SC-$CO_2$ mixture in high pressure technology is possible.

### 5.2. Extract Recovery from ILs with SC-$CO_2$ Process

The application of SC-$CO_2$ for recovering of products from ionic liquids relies principally on its solubility/phase behavior in the binary IL/$CO_2$ system. From the literature, behaviors of the IL/$CO_2$ system are different from other organic liquid/$CO_2$ systems. As an example, Blanchard et al. [137] studied the qualitative phase behavior of 1-*n*-butyl-3-methylimidazolium hexafluorophosphate ([bmim][PF$_6$])/$CO_2$ system within a wide pressure range. They studied the ([bmim][PF6])/$CO_2$ system phase behavior at 3100 bar and found two different phases. The two distinct phases were explained by the fact that at high-pressure density of pure $CO_2$, vapor phase increases but since the

liquid phase does not expand, the two phases will never become identical. Therefore, the IL/CO$_2$ system remains as two phases even at very high pressure; although the CO$_2$ solubility is quite high, the mixture will never become a single phase. Blanchard et al. in 2001 [137] showed that CO$_2$ dissolves in the liquid phase of ILs while the IL remains insoluble in the CO$_2$ vapor phase. In addition, Kroon et al. [138] experimentally studied the recovery of *N*-acetyl-(S)-phenylalanine methyl ester (APAM) from the ionic liquid 1-butyl-3-methylimidazolium tetrafluoroborate ([bmim][BF$_4$]) using SC-CO$_2$ as a cosolvent in extractions or as an antisolvent in precipitation [139]. These authors found that solubility of this APAM was lower in ionic liquid/SC-CO$_2$ mixture compared to the one in pure ionic liquid at room temperature and the extracted product has no trace of ionic liquid. Table 4 presents some ILs and their properties reported in literature used in SC-CO$_2$. Among the listed ILs, l-butyl-3-methylimidazolium bis-(trifluromethanesulflonyl)imide [C4mim][Tf$_2$N] has high thermal stability at 450.2 °C and density of 1.409 9 g·cm$^{-3}$ (at 25 °C with least viscosity of 44 (25 °C) MPa.s. The IL 1-*n*-octyl-3-methylimidazolium hexafluorophosphate ([C8-mim][PF$_6$]) is the most soluble in CO$_2$ at 60 °C with high viscosity of 866 at 20 °C.

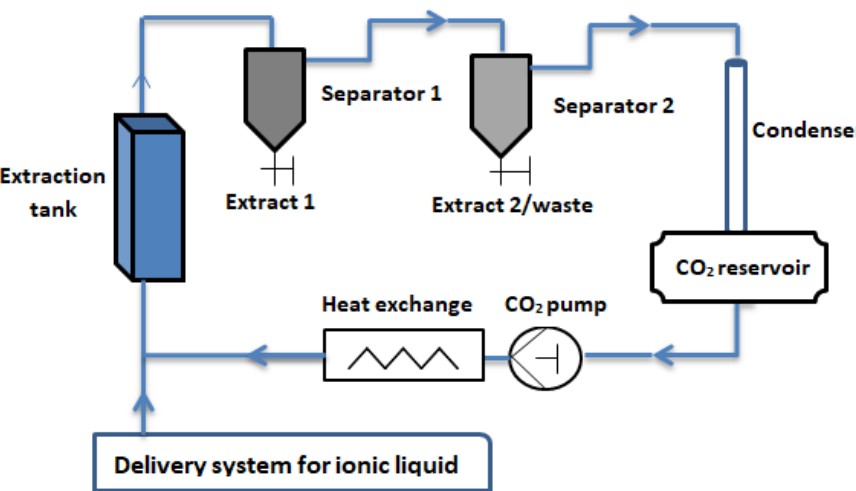

**Figure 3.** A simplified drawing of a process-scale IL/SC-CO$_2$ extractor.

**Table 4.** Properties and characteristics of ionic liquids used with supercritical $CO_2$ for extraction and separation purposes.

| Ionic Liquids | Thermal Stability/°C | Solubility in $CO_2$ | Viscosity/MPa.s | Density/g·cm$^{-3}$ |
|---|---|---|---|---|
| 1-butyl-3-methylimidazolium hexafluorophosphate ([bmim][PF$_6$]) | 410 [140] | 0.231 (40 °C, 14.17 bar); 0.236 (50 °C, 17.38 bar) 0.228 (60 °C, 15.79) [141] | 707 (293.15 K, 0.1 MPa) [142] | 1.345 (1 MPa, 323.15 K) [143] |
| 1-butyl-3-methylimidazolium tetrafluoroborate ([bmim][BF$_4$]) | 372.73 [143] | - | 122.35 [144] | 1.1647 [145] |
| 1-*n*-octyl-3-methylimidazolium hexafluorophosphate ([C8-mim][PF$_6$]) | - | 0.234 (40 °C, 17.93 bar) 0.223 (50 °C, 16.00 bar) 0.248 (60 °C, 17.38 bar) [141] | 866 (20 °C) | 1.211 (40 °C); 1.204 (50 °C); 1.197 (60 °C) [141] |
| 1-*n*-octyl-3-methylimidazolium tetrafluoroborate ([C8-mim][BF$_4$]) | 397 [144] | 0.197 (40 °C, 1 7.26 bar) 0.191 (50 °C, 15.61 bar) 0.160 (60 °C, 15.61 bar) [141] | 439 (20 °C) [146] | 1.080 (40 °C); 1.073 (50 °C); 1.066 (60 °C) [141] |
| 1-*n*-butyl-3-methylimidazolium nitrate ([bmim][NO$_3$]) | 297.49 [145] | 0.196 (40 °C, 15.47 bar) 0.169 ( 50 °C, 17.12 bar) 0.183 (60 °C, 18.37bar) [141] | 450.43 (283.15 K) [147] | 1.149 (40 °C); 1.143 (50 °C); 1.136 (60 °C) [141] |
| 1-ethyl-3-methylimidazolium ethyl sulfate ([emim][EtSO$_4$]) | - | 0.100 (40 °C, 16.43 bar) 0.103 (50 °C, 16.22 bar) 0.118 (60 °C, 14.36 bar) [141] | 97.2 (298.16 K) [148] | 1.225 (40 °C); 1.218 (50 °C); 1.213 (60 °C) [141] |
| *N*-butylpyridinium tetrafluoroborate ([*N*-bupy][BF$_4$]) | 373 [146] | 0.144 (40 °C, 15.47bar) 0.142 (50 °C, 16.57 bar) 0.166 (60 °C, 18.65 bar) [141] | 163.26 (298.15 K) [149] | 1.203 (40 °C); 1.197 (50 °C); 1.190 (60 °C) [141] |
| l-butyl-3-methylimidazolium bis-(trifluromethanesulflonyl)imide [C4mim][Tf$_2$N] | 450.2 [150] | - | 44 (25 °C) [145] | 1.409 9 (25 °C) [150] |

## 6. Conclusions

The significant potential of nonconventional methods as extraction and separation techniques for high-added value compounds from biomass or synthetic processes or fermentation broths in many process industries is increasingly being reviewed. The novel extraction technologies reviewed in this paper have the potential to significantly enhance the extraction yield. However, the maximum benefits of SCFE can be achieved only if a limitation regarding low concentration of compounds with high polarity and molecular weight is overcome. This review thus presented the introduction of ionic liquids as cosolvent in supercritical $CO_2$ extraction to overcome this limitation. Another promising technology recently reported that was not reviewed in this paper consists of gas assisted mechanical expression (GAME) which has shown interesting results on oil extraction from vegetable matrices. Future research priorities in the area of high pressure processing technology should concentrate on the use of ionic liquids as cosolvent as this has been proved to work in industrial scale for extraction/separation of high-added value compounds.

**Author Contributions:** G.F. hosting by T.V.O. and T.N.M. All authors have read and agreed to the published version of the manuscript.

**Funding:** This research was funded by the Cape Peninsula University of Technology of the Republic of South Africa.

**Conflicts of Interest:** The authors declare no conflict of interest.

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
