# Peer review of "Applications of Nonconventional Green Extraction Technologies in Process Industries: Challenges, Limitations and Perspectives"

_sustainability, doi:10.3390/su12135244_

Round 1

Reviewer 1 Report

The authors of the manuscript entitled "Applications of non-conventional “green” extraction technologies in process industries: challenges, limitations and perspectives"  describe the different friendly environmental extraction methods with focus in the use of ionic liquids in a supercritical fluid extraction method.  The idea is important but the form is a little confused mainly the English not flow. It is like ideas and ideas and no transitions in between.  More description is necessary in each method. and for example ionic liquids are used in other listed methods? the authors did't explain or highlight. 

In the attached file are my comments. But at this point the manuscript needs work on English language and is no at the level of sustainability. It can be reconsider with an English edit and more detailed descriptions

Author Response

Reviewer 1

Comment 1

The idea is important but the form is a little confused mainly the English not flow.

Response: We revised the English language of the manuscript by sending it off for language editing and revision.

Comments 2

It is like ideas and ideas and no transitions in between.  More description is necessary in each method. and for example ionic liquids are used in other listed methods? the authors did't explain or highlight.

Response:

We have expanded in more details the description of the highlighted technologies {see the beginning of each method}

Yes, ILs are used in other listed methods and this is one of the references [Molecules 2010, 15, 2405-2426] reviewing ionic liquids used in other listed methods. But this review aims to expose limitations of other methods even SCFE in different industrial application and show how ILs can be also used in SCFE to overcome his limitations since there are reviews on ILs in other methods already. That is why I did not highlight ILs in other techniques again.

We have also mentioned briefly other supercritical fluids apart from CO2 such as ammonia N2O, CHF3 and SF6 and their conditions. {see page 9 Line 250 to 262}

We have also included industrial examples of where the technologies can be applied in Table 3 as suggested {see Table 3 page 12}.

We have included  in Table 3, which explains the uses and characteristics of the different ionic liquid will added value {See Table 4 on Page 14}

Reviewer 2

Comment 1

General comments: there are some grammar errors; English should be improved, and some sentences should be checked; Check all the manuscript. 

Response: All the manuscript was revised accordingly. The English language was also revised by sending it off for language editing and revision.

Comment 2

The name of the plant species must be in italics in table and MS;

Response: We have corrected {see revised manuscript}

Comment 3

Check the reference style in the manuscript and in the list of references. more references of the recent years should be used.

Response: We have done it {see the revised manuscript, word file}

Comment 4

The MS must give more emphasis to the aim of this review. MS seems to be a list of information, no comparison with conventional extraction techniques was reported in terms of compounds and sustainability; the comparison is very important to understand the effictinevess and the added-value of the green extraction in process industries.

Response:

The aim of this review is well stated in the introduction of {page 2, line 52 to 58}

All the techniques reported in this manuscript are green extraction methods (which are also called non-conventional methods) and the MS is not comparing conventional with non-conventional but, presenting applications of non-conventional methods with their limitations in different industries.

Reviewer 2 Report

Authors of the manuscript entitled "Applications of non-conventional “green” extraction technologies in process industries: challenges, limitations and perspectives" describe the main uses of 5 green extraction techniques. The manuscript could fit with the aim of the journal after major revision; in my opinion the manuscript should be reconsidered after major revision. 

I have some suggstions to improve the content of the manuscript. 

General comments: there are some grammar errors; English should be improved, and some sentences should be checked; Check all the manuscript.  The name of the plant species must be in italics Iin tabke and MS; check the reference style in the manuscript and in the list of references. more references of the recent years should be used.

The MS must give more emphasis to the aim of this review. MS seems to be a list of information, no comparison with conventional extraction techniques was reported in terms of compounds and sustainability;  the comparison is very important to understand the effictinevess and the added-value of the green extraction in process industries. 

Author Response

Reviewer 2

Comment 1

General comments: there are some grammar errors; English should be improved, and some sentences should be checked; Check all the manuscript. 

Response: All the manuscript was revised accordingly. The English language was also revised by sending it off for language editing and revision.

Comment 2

The name of the plant species must be in italics Iin table and MS;

Response: We have corrected {see revised manuscript}

Comment 3

Check the reference style in the manuscript and in the list of references. more references of the recent years should be used.

Response: We have done it {see the revised manuscript, word file}

Comment 4

The MS must give more emphasis to the aim of this review. MS seems to be a list of information, no comparison with conventional extraction techniques was reported in terms of compounds and sustainability; the comparison is very important to understand the effictinevess and the added-value of the green extraction in process industries.

Response:

The aim of this review is well stated in the introduction of {page 2, line 52 to 58}

All the techniques reported in this manuscript are green extraction methods (which are also called non-conventional methods) and the MS is not comparing conventional with non-conventional but, presenting applications of non-conventional methods with their limitations in different industries.

Round 2

Reviewer 1 Report

Dear Authors,

The revised version of the manuscript has improved greatly, still a little inconsistencies and minor things. in the attached file.

The main problem is that the number in the citations sometimes don't correspond with  those in the reference section. Please recheck carefully that It is a big deal.

The authors must change that ad should be accepted

Author Response

The revised version of the manuscript has improved greatly, still a little inconsistencies and minor things. in the attached file.

The main problem is that the number in the citations sometimes don't correspond with those in the reference section. Please recheck carefully that It is a big deal.

The authors must change that ad should be accepted

Response: Please find attached file all the responses to your comments

Reviewer 2 Report

Manuscript was improved and it could be accepted in the present form.

Author Response

Thank you very much for your time. Thank you also for your comments and suggestions, they really helped me to improve the manuscript.